# Chromatin Accessibility Is Associated with Artemisinin Biosynthesis Regulation in *Artemisia annua*

**DOI:** 10.3390/molecules26041194

**Published:** 2021-02-23

**Authors:** Limeng Zhou, Yingzhang Huang, Qi Wang, Dianjing Guo

**Affiliations:** 1State Key Laboratory of Agrobiotechnology, School of Life Science, The Chinese University of Hong Kong, Hong Kong 999077, China; zhoulm1993@163.com (L.Z.); huangyingz@outlook.com (Y.H.); 2Artemisinin Research Center, Guangzhou University of Chinese Medicine, Guangzhou 510000, China; wangqi2019@gzucm.edu.cn

**Keywords:** *Artemisia annua*, glandular trichome, chromatin accessibility, ATAC-seq, artemisinin

## Abstract

Glandular trichome (GT) is the dominant site for artemisinin production in *Artemisia annua*. Several critical genes involved in artemisinin biosynthesis are specifically expressed in GT. However, the molecular mechanism of differential gene expression between GT and other tissue types remains elusive. Chromatin accessibility, defined as the degree to which nuclear molecules are able to interact with chromatin DNA, reflects gene expression capacity to a certain extent. Here, we investigated and compared the landscape of chromatin accessibility in *Artemisia annua* leaf and GT using the Assay for Transposase-Accessible Chromatin using sequencing (ATAC-seq) technique. We identified 5413 GT high accessible and 4045 GT low accessible regions, and these GT high accessible regions may contribute to GT-specific biological functions. Several GT-specific artemisinin biosynthetic genes, such as *DBR2* and *CYP71AV1*, showed higher accessible regions in GT compared to that in leaf, implying that they might be regulated by chromatin accessibility. In addition, transcription factor binding motifs for MYB, bZIP, C2H2, and AP2 were overrepresented in the highly accessible chromatin regions associated with artemisinin biosynthetic genes in glandular trichomes. Finally, we proposed a working model illustrating the chromatin accessibility dynamics in regulating artemisinin biosynthetic gene expression. This work provided new insights into epigenetic regulation of gene expression in GT.

## 1. Introduction

Malaria, as one of the most-deadly diseases, threatens the lives of nearly half of the world population. According to the World Health Organization (WHO), more than 200 million malaria cases and 405,000 deaths were reported in 2018 (WHO, World Malaria Report 2019). Artemisinin, a sesquiterpene metabolite discovered from Chinese herb *Artemisia annua*, is a potent drug against malaria [1,2]. Artemisinin-based combination therapies (ACTs) have been highly recommended since 2008 [3]. More recently, *Artemisia annua* has gained increasing attention as a potential source as a drug against COVID-19 [4,5]. However, despite its importance, artemisinin abundance in *Artemisia annua* is too low (about 0.1–1.0% dry weight) to meet the increasing demands [6,7]. Although genetically engineered yeast and tobacco have been used to produce artemisinin in vitro [8,9], *Artemisia annua* plants still remain the major commercial source for artemisinin production. There are also multiple breeding efforts for high-yielding *Artemisia annua*. For example, artemisinin-related traits were investigated and improved through population genetics [10,11] and molecular genetic breeding [12,13]. A comprehensive understanding of artemisinin’s biosynthetic regulatory mechanism is of great importance for improving the artemisinin yield through genetic engineering approach.

Glandular trichome (GT) is the dominant site for artemisinin biosynthesis and sequestration [14,15,16], although non-GT cells in leaf can also produce artemisinin [17]. Glandular trichome in *Artemisia annua* is a specialized 10-celled biseriate structure, composed of two non-photosynthetic apical cells, four photosynthetic subapical cells, two stalk cells, and two basal cells. The apical cells are surrounded by subcuticular space for the storage of secondary metabolites [18]. The artemisinin biosynthetic pathway originates from isopentenyl diphosphate (IPP), which is derived from either the mevalonate pathway or non-mevalonate pathway [19]. CYP71AV1, ADS, DBR2, and ALDH1 are the key enzymes involved in artemisinin biosynthesis [20,21,22,23,24]. The last biosynthetic step occurs in subcuticular space, where the dihydroartemisinic acid is converted to artemisinin in a photo-oxidation manner [25].

Artemisinin biosynthesis is subject to a sophisticated network incorporating regulatory transcription factors (TFs) that mainly belong to the MYB (myeloblastosis) family [6,26], WRKY (WRKY domain containing) family [27,28], AP2/ERF (AP2/ERF domain containing) family [29,30], bZIP (basic leucine zipper) family [31], and bHLH (basic helix-loop-helix) family [32]. For example, AaERF1 and AaERF2 are two positive regulators of artemisinin biosynthesis via binding to CBF2 and RAA motifs present in both *ADS* and *CYP71AV1* promoters in response to jasmonates [33]. Likewise, AaMYB1 positively regulated the expression of critical artemisinin biosynthetic genes and its overexpression resulted in increased artemisinin yield [26].

Chromatin accessibility is the degree to which nuclear macromolecules are able to make physical access to DNA, reflecting the gene regulatory capacity [34]. The landscape of chromatin accessibility changes dynamically in response to both external stimuli and developmental cues [35,36]. It is mainly determined by the occupancy and topological organization of nucleosomes, together with other chromatin binding factors. Nucleosome is the core structural unit of chromatin assembly in eukaryotic species. An octamer of histone proteins is encircled by about 147 bp of DNA [37,38]. However, nucleosomes are not uniformly organized across the genome. While densely arranged at facultative and constitutive heterochromatin, histones are depleted or loosely placed at regulatory loci, such as enhancers and promoters [39]. Transcription factor (TF) is an important player in DNA accessibility modulation. By dynamically competing with histones, TFs rearrange nucleosome occupancy and increase DNA accessibility. Mutually, the accessibility landscape of a cell type in turn adjusts TF binding [34]. Therefore, chromatin accessibility mirrors both aggregate TF binding events and the regulatory potential of a genetic locus.

Chromatin accessibility is generally detected by profiling the susceptibility of chromatin to either enzymatic methylation or cleavage of its constituent DNA [34]. There are several methods for DNA accessibility measurement, such as DNase-seq [40], ATAC-seq (assay for transposase-accessible chromatin using sequencing) [41], and MNase-seq [42]. ATAC-seq takes advantage of a hyperactive Tn5 transposase which cuts accessible DNA regions and meanwhile adds Illumina adaptor to the ends. Due to the high efficiency of Tn5-mediated adaptor ligation, ATAC-seq thus required less sample input amount and a simplified library construction procedure as well [41].

Previous investigations on artemisinin biosynthetic regulation were mostly focused on TFs that were specifically expressed in GT, whereas the epigenetic regulation of gene expression in *Artemisia annua* remains largely unknown. In this study, we analyzed and compared the chromatin accessibility of *Artemisia annua* GT and leaf using ATAC-seq [41], aiming to provide new insights into the regulatory mechanism of artemisinin biosynthesis from an epigenetic perspective.

## 2. Results

### 2.1. Mapping Accessible Chromatin in Artemisia annua

We sought to investigate the accessible chromatin in GT and leaf of *Artemisia annua*. Glandular trichomes from the young leaves of six-month-old plants were isolated with glass beads and manually collected under a microscope. Isolated GTs contained intact apical and sub-apical cells (stalk and basal cells were missing) and little contamination (Figure 1A). Meanwhile, GTs were removed by gentle brushing using a writing brush to minimize GT in leaf sample. The autofluorescence signal in GT under 488 nm excitation channel was used as an indicator for efficient GT removal. Only a few GTs remained in the leaf sample after brushing and thus were neglectable for ATAC-seq and RNA-seq library construction (Figure 1B,C, Appendix A).

Spearman correlation analysis showed that the ATAC-seq libraries displayed good reproducibility for both tissues (Figure 2A). In total, we identified 60,823 and 58,422 ACRs (Accessible Regions) in GT (two replicates) and 58,387 and 58,452 ACRs in leaf (two replicates) (Figure 2B). Overall, the expression level of genes with ACRs was significantly higher than those without ACRs in both GT and leaf (*p* < 2.2×10^−16^
Figure 2C), indicating that ACRs may be involved in promoting the expression of their associated genes. In addition, more than half of the ACRs appeared to be located at distal regions (Figure 2D) and the distance between distal ACRs and transcription start site (TSS) was about 10 kb (Figure 2E). This is consistent with the genome features of large-genome plant species [43].

### 2.2. Glandular Trichome High-Accessible Differential Accessible Regions (DARs) Are Likely Involved in Regulating GT-Specific Cellular Function

We identified 5431 and 4045 differential accessible regions (DARs) with high and low accessibility in GT, compared to leaf (fold change ≥ 2, *q* value < 0.05, Figure 3A, Appendix A). These regions were assigned to 4869 and 3231 genes by associating with the nearest genes. It is noteworthy that a higher proportion of proximal and genic ACRs were found in GT high-accessible DARs. Contrarily, for GT low-accessible DARs, the majority of ACRs lay in distal regions (Figure 3B). Proximal and genic ACRs were more likely to impose a direct and profound effect on transcriptional regulation. For genes only associated with distal ACRs, their expression levels were significantly lower than those associated with proximal ACRs (Figure 3C). In addition, we found that additional distal ACR exhibited little effects on genes with proximal ACRs (Figure 3C).

In total, we identified 5553 and 4865 differential expressed genes (DEGs) that were upregulated and downregulated in GT respectively, and we designated them as GT-up and GT-down DEGs (fold change ≥ 2, *q* value ≤ 0.05, Figure 3D, Appendix A). We found that 728 GT-up DEGs were overlapped with GT high-accessible DARs (P(X ≥ 728) = 1.51×10^−46^, hypergeometric test, Figure 3D, Appendix A), while only 205 GT-down DEGs were overlapped with GT low-accessible DARs. We speculated that the overlapped gene sets may be associated with GT-specific biological function regulated by chromatin accessibility. In GO (Gene Ontology) enrichment analysis, the 728 overlapped genes were enriched in biological processes such as “metabolic process”, “photosynthesis, light harvesting”, “biosynthetic process”, etc. (Figure 4, Appendix A). *Artemisia annua* glandular trichomes are known to be active in photosynthesis [44] and bio-factories for the production of artemisinin and other secondary metabolites. The top ranked GO terms identified for molecular function were “binding”, “transporter activity”, and “catalytic activity”, which can be linked to active biosynthesis, transport, and secretion function of GTs (Figure 4, Appendix A). Plasmid, cytoplasm, and chloroplast were the top three ranked subcellular sites for these overlapped genes (Figure 4, Appendix A).

### 2.3. Some Artemisinin Pathway Gene Expressions Are Associated with Chromatin Accessibility

Next, we investigated the possible association between artemisinin biosynthesis and chromatin accessibility. Notably, several GT-specific artemisinin biosynthesis genes: *HMGR* (AA201470), *DXS* (AA422860), *DXR* (AA444110), *CYP71AV1* (AA566140), and *DBR2* (AA049700), showed higher DNA accessibility in GT compared to that in leaf (Figure 5A). The gene expression and chromatin accessibility were further validated by RT-qPCR (Real-time quantitative PCR) and ATAC-qPCR (Assay for Transposase-Accessible Chromatin using quantitative PCR) (Figure 5B). HMGR, DXS, and DXR are involved in the biosynthesis of FPP (farnesyl pyrophosphate), the necessary precursor for terpene [45], while CYP71AV1 (AA566140) and DBR2 (AA049700) are two key enzymes in artemisinin biosynthesis and actively expressed in GT [22,23]. Interestingly, some artemisinin pathway genes, such as *HDR* (AA477650), *HMGS* (AA036900 and AA288120), and *HMGR* (AA271980), did not show GT high-accessible DARs or associated ACRs. Also, several GT-up DEG genes showed no difference in chromatin accessibility between leaf and GT. For example, a GT-up DEG *CYP71AV1* (AA502080, not the functional one), a homolog of *CYP71AV1* (AA566140, the functional one) [6], showed no difference in ACR, which is possibly related with its low expression level (Figure 5C, Appendix A). Taken together, the association between chromatin accessibility and artemisinin biosynthesis gene expression is highly complex. However, our results indicated that at least some of the key pathway genes are likely regulated by chromatin accessibility. In *Artemisia annua*, artemisinin analogs include artemisinic acid, dihydroartemisinic aldehyde, and arteannuin B. The three all share the same biosynthetic pathway with artemisinin, therefore, the regulation of accessible regions on genes associated with artemisinin analogs’ biosynthesis is consistent with artemisinin.

Putative cis-responsive elements (CREs) and TF families that may play a role in modulating the GT-specific transcriptome were identified from the GT high DARs (Appendix A). We focused on motifs with a *p* < 1×10^−10^ for the enrichment analysis. The top four overrepresented TF families were MYB, AP2, bZIP, and C2H2 family (Figure 6A), which were all enriched in the ACR center (Figure 6B). The five potentially DAR-regulated artemisinin biosynthetic genes (Figure 5A) harbored motifs for at least two TF families in their ACRs (Figure 6C). Transcription factors that are potentially involved in regulating the observed DARs can be identified by asking whether they are differentially expressed and meanwhile putatively bind to differentially accessible DNA [34]. We thus constructed a regulatory network by incorporating the identified motifs and TFs with high expression (GT/leaf fold change ≥ 2, TPM ≥ 50, Figure 6D). From the constructed network, six MYB, one C2H2, three AP2, and three bZIP transcription factors were selected as potential regulators (Figure 6C,D, Appendix A). These TFs showed significantly high expression levels in GT and were very likely involved in artemisinin biosynthesis regulation since their binding motifs were discovered in the accessible regulatory regions of several key artemisinin biosynthetic genes.

Based on these findings, chromatin accessibility seems to play an important role in regulating gene function in GT. We further proposed a working model to describe the possible association between chromatin accessibility and tissue-specific artemisinin biosynthesis gene expression in *Artemisia annua* (Figure 6E). In GT, the highly accessible regulatory regions for several artemisinin biosynthetic pathway genes (e.g., *CYP71AV1* and *DBR2*) provide access for TF binding, which consequently promotes gene expression. Contrarily, the counterpart regions in leaf were less accessible, resulting in low gene expression.

## 3. Discussion

Glandular trichome and leaf cells exhibit different morphology and functions, which is a readout of cell-specific gene expression pattern during differentiation. According to our analysis results, one-sixth of the total genes were differentially expressed in GT compared to leaf. These genes are likely responsible for the establishment and maintenance of cellular identity and function of glandular trichome, which is the main site for secondary metabolite biosynthesis (Figure 3D). Plant gene expression regulation is a complex process subject to precisely coordinated multilayer controls. Chromatin accessibility is considered as a context-dependent regulatory mechanism, which partially reflects gene expression potential [34]. In our study, we found that GT high DARs overlapped with GT-up DEGs, suggesting a possible link between chromatin accessibility and gene expression. Glandular trichome high DAR greatly contributes to GT-specific biological function. Specifically, the GT high DAR-associated genes showed an enrichment in metabolism-related processes, consistent with GT’s role as the bio-factories for the production of artemisinin and other plant secondary metabolites. We found that five of the GT-specific artemisinin synthesis pathway genes were associated with higher chromatin accessibility and tended to show generally higher expression levels compared to those that were not subject to ACR regulation. Chromatin remodeling complex is involved in the regulation of chromatin landscape. In *Tetrahymena thermophila*, ChIP-Seq (chromatin immunoprecipitation assays with sequencing) analysis of a bromodomain-containing protein indicated that it primarily binds to highly expressed genes during growth [46]. Likewise, in yeast, chromatin remodeling complex SWI/SNF and RSC occupancies are greatest at the most highly expressed genes [47]. Another chromatin remodeler, Ino80C in yeast, also preferred to target highly expressed genes [48]. It is reasonable to speculate that highly expressed genes in GT, such as *CYP71AV1* (AA566140), are more likely subject to the regulation of chromatin remodeling and thus show greater DNA accessibility plasticity between different tissues.

Interestingly, photosynthesis and primary metabolites were also highly enriched in GT-up DEG and GT high DAR overlapped genes in GO analysis. *Artemisia annua* GTs contain photosynthetically active chloroplast, and photosynthesis-related proteins are among the top 10 most highly expressed transcripts [44,49]. Photosynthetic activity was also reported in tobacco glandular trichome, which has particular Rubisco type uniquely adapted to secretory cells, where CO_2_ is released by the active specialized metabolism [50]. We presumed that the energy and substrate produced by photosynthesis better fuel and enhance the active metabolic processes in GT. However, the regulation mechanism of highly active photosynthesis is largely unknown. Our results indicated that they were possibly associated with greater chromatin accessibility. Genes involved in photosynthesis showed greater ACRs and higher transcription in GT, compared to those in leaf. The concurrently increased transcription and enhanced chromatin accessibility have been reported in root cell identity maintenance [35], response to plant hormone [36], and flower development [51], etc. Taken together, these findings further emphasized the importance of chromatin accessibility in transcriptional regulation throughout plant development and growth.

Our study pointed out a possible link between chromatin accessibility and gene expression in GT. However, among these GT high DARs, only 728/4869 were associated with GT-up DEGs. Similarly, another study in *Arabidopsis thaliana* also reported that only 851 out of 3282 root hair-up DEGs were overlapped with root hair high DARs [35]. In our study, ADS (AA425620), a key enzyme for artemisinin biosynthesis, was highly expressed in GT but no nearby ACR was detected. This discrepancy of DEG and DAR was actually common and has been reported in other biological processes, such as flower bud induction in *Arabidopsis thaliana* and early embryo development in *Oryzias latipes* [51,52]. One possible explanation is the dynamic chromatin remodeling event during gene regulation. In fact, some chromosomal regulatory regions become accessible before significant increase of transcripts [51]. The discrepancy also reflects the complexity of transcriptional regulation, which is well-orchestrated in a complex, multi-layered, and interconnected fashion [34]. Due to the large genome size (1.74 Gb), high heterozygosity (1.0–1.5%), highly repetitive sequence (>60%), and low GC content (31.5%), *Artemisia annua* genome assembly needs further improvement, despite the efforts dedicated to accomplishing the current version (approximate 40,000 scaffolds) [6]. The poorly assembled genome may cause mis-assigned ACRs. In addition, some ACRs function as long-distance enhancers involved in chromatin looping [53], which may result in a high proportion of distal ACRs (Figure 1C). With respect to the relationship between gene expression and chromatin accessibility, the gene expression level used in this study reflects the total mRNA amount without taking its dynamics and stability into consideration. Methods that measure the production of nascent RNAs, such as GRO-seq (Global run-on sequencing), may be more precise to study the effect of chromatin accessibility on gene expression [54].

In conclusion, we comprehensively investigated and compared the landscape of chromatin accessibility in GT and leaf of the important medicinal plant *Artemisia annua*. Our results further validated the complex gene expression regulation in artemisinin biosynthesis and provided new insights into the regulatory mechanism of GT-specific biological function from an epigenetic perspective.

## 4. Materials and Methods

### 4.1. Plant Materials and Sample Preparation

*Artemisia annua* seeds (a high artemisinin-producing chemotype, artemisinin content: 1.34% for dry weight) were purchased from Youyang, Sichuan, China. *Artemisia annua* plants were grown under natural light condition in the greenhouse of the Chinese University of Hong Kong.

Young and just expended leaves (the fourth to sixth leaf from the top on the stem) of 6-month-old plants were collected into 50 mL sterile Falcon tubes containing 20 mL PBS (Phosphate buffered saline) with 0.1% TritonX-100 and 5 mL 0.5 mm glass beads (BioSpec Products, Bartlesville, OK, USA). The tube was shaken by hand vertically and vigorously more than 300 times to abrade GT off the leaves to obtain crude GTs [44]. Then, individual GT was manually collected under microscopy with a home-made tool (made of syringe, infusion tube, and capillary tube).

For the leaf sample, leaves at the same age as leaves used for GT preparation were collected. As many GTs as possible were removed by gently brushing with a writing brush. After brushing, leaves were checked under a fluorescence microscope to confirm the removal of trichomes. Purified GTs and leaves were directly used for ATAC-seq and RNA-seq library construction.

### 4.2. ATAC-seq Library Generation, Sequencing, and Mapping

There are several methods for DNA accessibility measurement, such as DNase-seq [55], ATAC-seq [41], and MNase-seq [42]. Here, we used ATAC-seq to measure the accessible regions in *Artemisia annua* since the GT sample could not meet the input requirements for other methods. ATAC-seq was performed as previously described with minor modifications [41,43]. Glandular trichomes and leaves were finely ground in liquid nitrogen. 1 mL (for GT) and 3 mL (for leaf) prechilled lysis buffer (15 mM Tris-HCl, pH 7.5, 20 mM NaCl, 80 mM KCl, 0.5 mM spermine, 5 mM 2-mercaptoethanol, 0.5% TritonX-100, 1 × proteinase, and 1 × PMSF(phenylmethylsulfonyl fluoride)) were added to the ground sample and mixed by pipetting. The slurry was filtered twice through a 40 μm filter and centrifuged at 1000 g at 4 °C for 10 min. The supernatant was discarded and the pellet was washed twice with lysis buffer and once with tag buffer (10 mM Tris-HCl, pH 8.0, 5 mM MgCl_2_). The nuclei were incubated with 2 μL Tn5 transposase (Vazyme, Nanjing, China, TD501) in 40 μL tag buffer at 37 °C for 30 min without rotation. The integration products were purified using DNA clean and concentration column (Zymo, Orange County, CA, USA) and then amplified using non-hot-start DNA polymerase for 10–13 cycles. Amplified libraries were purified using VAHTS DNA clean beads (Vazyme, Nanjing, China) to remove adapters.

ATAC-seq libraries were pooled and sequenced using the Illumina Hiseq X Ten platform. Raw reads were trimmed using trim_galore with parameter: “-q 25 --phred33 --length 40 --paired”. Trimmed reads were aligned to *Artemisia annua* reference genome using bowtie2 with the following parameters: “-p 10 -X 1000” [56]. Aligned reads were sorted, followed by PCR duplicates and multiple aligned reads’ removal with sambamba [57].

### 4.3. Identification and Annotation of ACRs and Differential Accessible Regions (DARs)

ACRs were identified using MACS2 (Model-based Analysis of ChIP-seq) with parameter:“ --nomodel --shift 100 --extsize 200” [58]. Peak files were merged and reads in each ACR were counted with Feature-Counts and DARs (differential accessible regions) were detected with DEseq2 [59]. ACRs were annotated with ChIPseeker and the *Artemisia annua* genome annotation file [60].

### 4.4. GO Analysis

Since we did not find GO numbers in the *Artemisia annua* genome annotation file, BLAST (Basic Local Alignment Search Tool) was used to search *Artemisia annua* ortholog genes from sunflower *Helianthus annuus* genome. DAR-associated genes were transferred to their ortholog genes in sunflower *Helianthus annuus* to perform GO analysis with online tools (http://geneontology.org/, accessed on 19 February 2021).

### 4.5. RNA-seq Library Generation, Sequencing, and Mapping

Total RNA of GT and leaves were extracted by TRIzol (Invitrogen, Waltham, MA, USA) and mRNA was purified by the Dynabeads mRNA DIRECT kit (Invitrogen). First-strand DNA was synthesized with reverse transcription with oligo dT as a primer and second-strand DNA was synthesized by random hexamer and followed by Tn5 tagmentation. DNA fragments with adapters were amplified and pooled for sequencing. Raw reads were trimmed using trim_galore with parameter: “-q 25 --phred33 --length 40 --paired”. Trimmed reads were aligned to the *Artemisia annua* reference genome using Hisat2 [61]. Aligned reads were sorted with sambamba [57] and clean reads were used to calculate TPM (Transcripts Per Million) and DEGs with DEseq2 [59].

### 4.6. RT-qPCR and ATAC-qPCR

RNA was extracted with TRIzol reagent (Invitrogen,), followed by reverse transcription with SuperScript IV reverse transcriptase (Invitrogen). cDNA was then used for RT-qPCR (Vazyme). Beta-actin was used as an internal control. Primers used for RT-qPCR are listed in Appendix A.

ATAC-qPCR was performed using the ATAC-seq library with 1 ng DNA as a DNA sample. Values was first normalized by ACT1 (ACTIN1) and then by genomic DNA. ACT1 was selected for its low variation among different samples and high accessibility. Primers used for ATAC-qPCR are listed in Appendix A.

### 4.7. Network Construction

Motif enrichment was performed with Homer using default parameters [62]. Motif discovery was performed with FIMO (Find Individual Motif Occurrences) in MEME suite (Motif-based sequence analysis tools) using default parameters [63]. Transcription factors were selected according to their expression levels (TPM > 50). The network was plotted with Cytoscape [64].

## Figures and Tables

**Figure 1 molecules-26-01194-f001:**
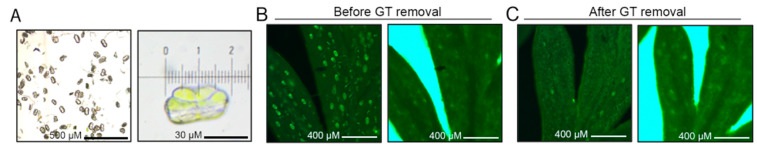
Sample preparation for GT (glandular trichome) and leaf for ATAC-seq and RNA-seq. (**A**) Purified GTs (left). Scale bar: 500 μM. A zoom-in view of the purified GT (right). Scale bar: 30 μM. (**B**,**C**) *Artemisia annua* leaf under fluorescence (left) and light (right) microscopy before (**B**) and after (**C**) trichome removal. Scale bar: 400 μM.

**Figure 2 molecules-26-01194-f002:**
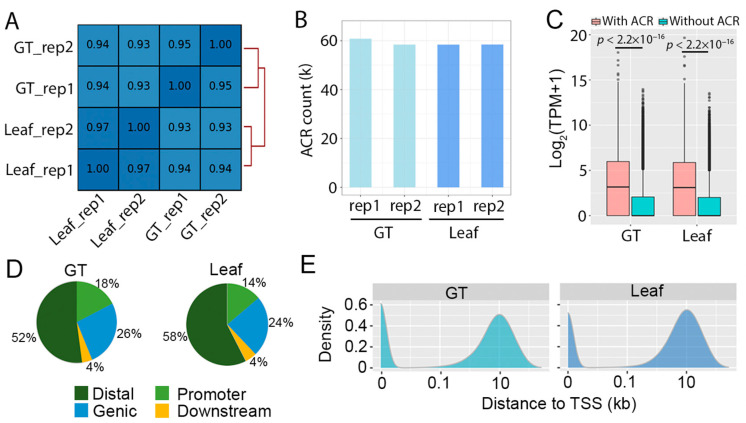
ATAC-seq analysis of GT and leaf in *Artemisia annua.* (**A**) Heatmap and clustering of Spearman correlation of ATAC-seq libraries. (**B**) ACR (accessible region) number detected in GT and leaf samples with MACS2. (**C**) Expression level of genes with or without associated ACRs in GT and leaf. The statistical analysis was performed using Student’s *t*-test. (**D**) Distribution of ACR within different genomic regions in GT and leaf. (**E**) Density of distance from ACR to TSS (transcriptional start site) in GT and leaf.

**Figure 3 molecules-26-01194-f003:**
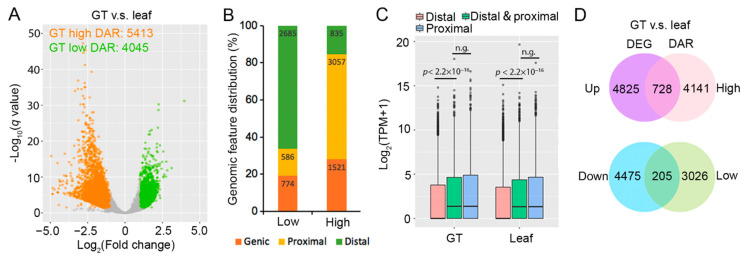
Chromatin accessibility in GT played a more important role in gene expression regulation. (**A**) Volcano plot of differential accessible regions (DARs) identified by DESeq2. The orange and green circles represent differentially accessible regions meeting the threshold (*q* value ≤ 0.05 and fold change of ≥ 2). (**B**) Genomic feature distribution for GT low and GT high DARs. Proximal and genic regions were enriched in GT high DARs. (**C**) Expression level of genes associated with distal ACR, distal and proximal ACR, and proximal ACR. Genes associated with proximal ACRs showed higher gene expression levels. The statistical analysis was performed using Student’s *t*-test. “n.g.” means not significant. (**D**) Venn diagram of overlaps between DEG (purple and blue stands for GT-up and GT-down DEGs, respectively) and DAR (pink and green stands for GT high and GT low DARs, respectively) associated genes.

**Figure 4 molecules-26-01194-f004:**
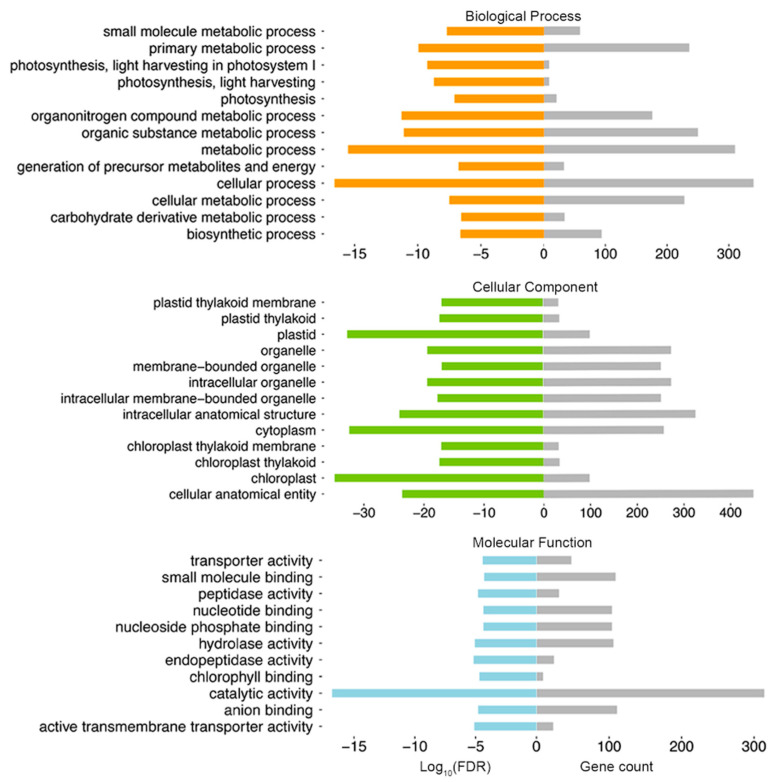
Gene Ontology (GO) enrichment analysis of the overlapping genes between GT-up DEGs and GT high DAR-associated genes (number of genes = 728). The fold of differential enrichment (Log_10_FDR (false discovery rate)) associated with each GO term classified under biological process (BP), cellular component (CC), and molecular function (MF) is represented in orange, green, and blue respectively, on the left half of the bar plots. The number of genes associated with each of these GO terms is shown in grey on the right half of the bar plots. The full GO enrichment results are listed in Appendix A.

**Figure 5 molecules-26-01194-f005:**
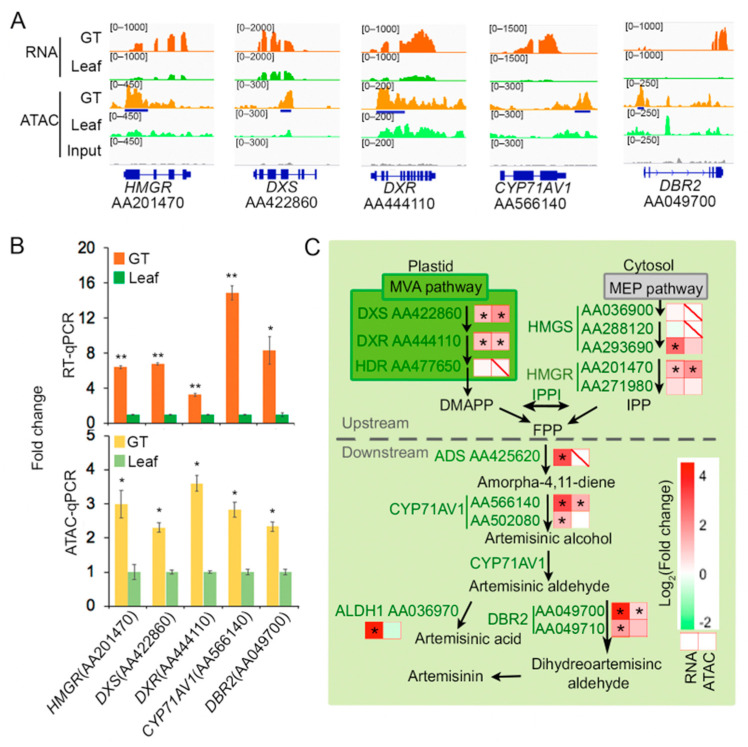
Chromatin accessibility plays a role in artemisinin biosynthesis regulation. (**A**) Integrated genome browser view of RNA-seq and ATAC-seq for five artemisinin biosynthetic genes that showed higher transcription level and higher chromatin accessibility in GT. Genomic DNA was used as a negative control. Blue bar indicated the DAR. (**B**) Results in (A) were further validified by RT-qPCR (Real-time quantitative PCR) and ATAC-qPCR (Assay for Transposase-Accessible Chromatin using quantitative PCR) For RT-qPCR, *beta-actin* was used as an internal control. For ATAC-qPCR, values were first normalized by internal control *ACT1* and then by genomic DNA. “*” means *p* ≤ 0.05, “**” means *p* ≤ 0.01. (**C**) Alterations of gene expression and ACRs in artemisinin biosynthetic pathway. Heatmaps show fold change of gene expression (left) and ACRs (right). “*” means *q* value ≤ 0.05 and fold changes ≥ 2. Backslash in right box means that no associated ACRs were detected for that gene. DXS, 1-deoxy-d-xylulose-5-phosphate synthase; DXR, 1-deoxy-Dxylulose-5-phosphate reductoisomerase; HDR, hydroxy-2-methyl-2-(*E*)-butenyl 4-diphosphate reductase; HMGS, 3-hydroxy-3-methyl-glutaryl coenzyme A synthase; HMGR, 3-hydroxy-3-methyl-glutaryl coenzyme A reductase; IPPI, isopentenyl pyrophosphate isomerase; ADS, amorpha-4,11-diene synthase; CYP71AV1, amorphadiene-12-hydroxylase; DBR2, artemisinic aldehyde ∆11(13) reductase; ALDH1, aldehyde dehydrogenase; MVA, mevalonic acid; MEP, 2-C-methyl-d-erythritol 4-phosphate; DMAPP, dimethylallyl diphosphate; FPP, farnesyl pyrophosphate; IPPI, isopentenyl pyrophosphate.

**Figure 6 molecules-26-01194-f006:**
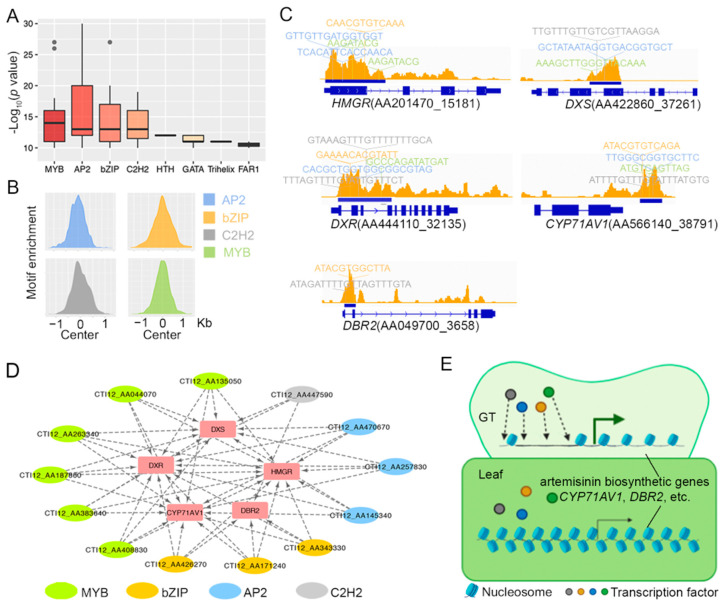
Regulatory network of artemisinin biosynthetic genes. (**A**) Motif enrichment in GT high DAR calculated with Homer. (**B**) AP2, bZIP, C2H2, and MYB TF binding motifs were enriched in GT high DAR center. (**C**) AP2, bZIP, C2H2, and MYB TF binding motifs were discovered by FIMO in the five artemisinin biosynthetic genes that were possibly regulated by DAR. (**D**) A gene regulatory network for artemisinin biosynthetic genes based on TF expression and binding motif detected. Transcription factors selected meet the threshold (GT/leaf fold change ≥ 2, TPM ≥ 50). Motifs detected in DAR build the potential link between the TF and DAR-associated artemisinin biosynthetic genes. (**E**) Working model illustrating the possible association between chromatin accessibility and tissue-specific artemisinin biosynthesis genes.

## Data Availability

The high-throughput sequencing data generated in this study were deposited in the NCBI SRA database with accession number PRJNA663216.

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
