# Peer review of "Chromatin Accessibility Is Associated with Artemisinin Biosynthesis Regulation in Artemisia annua"

_molecules, 2021, doi:10.3390/molecules26041194_

Round 1
Reviewer 1 Report
Zhou et al. presented and interesting report on the regulation of artemisinin biosynthetic and precursor supply genes via chromatin accessibility in Glandular Trichomes of A. annua. This is, up to my knowledge, the first study of that kind performed on A. annua. However, chromatin accessibility is a very common mechanism for regulation of gene expression within all living organisms, hence my lower notes for "Significance of Content" and "Interest to the readers".
Paper also requires minor modifications, including clarifications to the results and material and methods section as highlighted in the attached draft. Some citations missing as well. Conclusions are justified but better quality reference genome assembly would definitely improve analysis done by the authors.

Reviewer 2 Report
The authors presented on artemisinin, a type of sesquiterpene lactone, biosynthetic regulation. The content of this manuscript is well organized. This manuscript contains content that is of interest to experts in this field as well as non-experts. In particular, elucidation of the mechanism of biosynthetic regulation of artemisinin may help the development of therapeutic agents for infectious diseases such as COVID-19.
To make this manuscript ever better, please consider the following comments.
Add the structural formula of artemisinin to the text.
Add the biosynthetic pathway of artemisinin to the text.
In relation to the results obtained this time, the authors should mention whether the biosynthetic regulation of artemisinin analogs contained in Artemisia annua can be also considered in the same manner as artemisinin.
